# Aspect-Category Enhanced Learning with a Neural Coherence Model for Implicit Sentiment Analysis

**Jin Cui[1], Fumiyo Fukumoto[1], Xinfeng Wang[1], Yoshimi Suzuki[1],**
**Jiyi Li[1] and Wanzeng Kong[2]**

[1] University of Yamanashi, Kofu, Japan
[2] Hangzhou Dianzi University, Hangzhou, China

{g22dtsa5,fukumoto,g22dtsa7,ysuzuki,jyli}@yamanashi.ac.jp,
kongwanzeng@hdu.edu.cn

## Abstract

Aspect-based sentiment analysis (ABSA) has been widely studied since the explosive growth of social networking services. However, the recognition of implicit sentiments that do not contain obvious opinion words remains less explored. In this paper, we propose aspect-category enhanced learning with a neural coherence model (ELCoM). It captures document-level coherence by using contrastive learning, and sentence-level by a hypergraph to mine opinions from explicit sentences to aid implicit sentiment classification. To address the issue of sentences with different sentiment polarities in the same category, we perform cross-category enhancement to offset the impact of anomalous nodes in the hypergraph and obtain sentence representations with enhanced aspect-category. Extensive experiments on benchmark datasets show that the ELCoM achieves state-of-the-art performance. Our source codes and data are released at https://github.com/cuijin-23/ELCoM.

## 1 Introduction

Aspect-based sentiment analysis (ABSA) has been one of the major research topics in NLP since a large volume of reviews has been accessible via social networking services. Much of the previous work on ABSA has focused on explicit sentiment (Yang and Zhao, 2022; Yan et al., 2021; Chen et al., 2022), while implicit sentiment, in which obvious sentiment polarity words do not appear in the sentence but convey sentiments, often appears in reviews. As illustrated in Figure 1, "The food here is rather good, but only if you like to wait for it." in $s_1$, cannot be clearly identified as "negative" with respect to the aspect category "service," because it does not include any opinion words related to it. Even less work on implicit sentiment mainly leverages intrasentential information to exploit effective contexts, for example, syntactic information

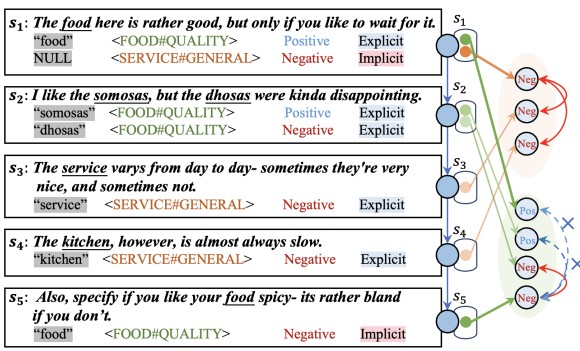

Figure 1: A review from SemEval-2016: Words in angle brackets refer to aspect-categories. The blue lines indicate that the review is coherent. The orange and green lines show that each "service" and "food" category points out its sentiment.

from dependency trees which results in insufficient representations capturing their contexts.

One feasible solution is to leverage one aspect of document quality: a document forms coherence if it is well-written and easy to understand. Many coherence models such as entity-based approaches (Jeon and Strube, 2022), and neural coherence models (Nguyen and Joty, 2017; Moon et al., 2019) have been proposed and applied to various NLP tasks. However, it is often the case that each sentence in a review includes several different aspect-categories, and even the same aspect-category within a review expresses different sentiment polarities. As illustrated in bold lines in Figure 1, $s_1$ has two different aspect-categories, "food" and "service," with different polarities. Likewise, "food" in $s_1$ is positive, while that of $s_5$ shows negative polarity. Much of the existing coherence models obtained from the distribution of entities or adjacent sentences are inadequate and make it hard to predict an accurate polarity of implicit sentiment.

Motivated by the issue mentioned above, we propose an aspect-category enhanced learning with a neural coherence model (ELCoM) that leverages coherence information for ABSA, especially ben-

eficial for implicit sentiment classification. On the one hand, observing from the SemEval-15 and SemEval-16 ABSA datasets, (i) more than 50% of the reviews had the same sentiment polarity regardless of different aspect-categories, and (ii) more than 70% of reviews had the same sentiment polarity if they had the same aspect-category. These observations indicate that (i) the reviewer's opinions are likely to be preserved throughout the review and (ii) the aspect-category is a strong clue for capturing sentence-level coherence to aid implicit sentiment classification. We thus utilize document-level coherence to deal with (i) and exploit the relationships among sentences and aspect-categories by utilizing a hypergraph for (ii).

On the other hand, the rest of the 20~30% reviews include opposite polarities with each other even in the same category, while these reviews preserve coherence. To compensate for the dilemma of the side effect, we perform cross-category enhancements in the hypergraph. More specifically, we utilize a self-attention (SA) filtering to offset the impact of anomalous nodes, apply retrieval-based attention (Rba) technique (Zhang et al., 2019) to learn the enhanced embedding of the aspect term, and finally obtain sentence representations with enhanced aspect-category.

The main contributions of our work can be summarized as follows: (1) we propose an ELCoM that learns document-level coherence by using contrastive learning and sentence-level by hypergraph for mining opinions to aid implicit sentiment classification; (2) we propose cross-category enhancements on node embedding to offset the impact of anomalous nodes to correctly identify the sentiment of the same categories but have different polarities; and (3) extensive experiments on SemEval-2015 and 2016 show that our method achieves state-of-the-art performance.

## 2 Related Work

### 2.1 Implicit Sentiment Analysis

To date, there has been very little work on implicit sentiment analysis. To address this issue, Li et al. (2021a) proposed a supervised contrastive pre-training model that learns sentiment clues from large-scale noisy sentiment-annotated corpora. Wang et al. (2022) established the causal representation of the implicit sentiment by identifying the causal effect between the sentence and sentiment. These attempts achieved better perfor-

mance, while their models treat a sentence independently and ignore how sentences are connected as well as how the entire document is organized to convey information to the reader.

In the context of leveraging a whole document for sentiment analysis, Chen et al. (2020) assumed intra- and inter-aspect sentiment preferences to classify aspect sentiment. Their approach is similar to ours since they focused on aspect-categories to capture sentiment polarity tendencies. However, their model only explored explicit sentiment classification. Cai et al. (2020) attempted implicit aspect-category detection and category-oriented sentiment classification by applying a hierarchical graph convolutional network. Their approach is also similar to ours in that they consider document-level information. The difference is that we leverage contextual features by utilizing both document- and sentence-level coherence based on aspect-categories to learn more fine-grained contextual representation.

### 2.2 Coherence Analysis

With the success of deep learning (DL) techniques, many authors have attempted to apply DL to learn features for coherence. One attempt is to model coherence as the relationship between adjacent sentences. This type includes the CNN (Nguyen and Joty, 2017), a hierarchical RNN (Nadeem and Ostendorf, 2018), and an attention mechanism (Liao et al., 2021). Another attempt is coherence as a whole document. This includes inter-sentence discourse relations (Moon et al., 2019) as well as word- and document-level coherence (Farag and Yannakoudakis, 2019). Our approach lies across both attempts and provides a comprehension framework for sentiment coherence.

Most of the attempts to apply neural coherence models to NLP tasks, such as text generation (Parveen et al., 2016; Guan et al., 2021), summarization (Eva et al., 2019; Goyal et al., 2022), and text quality assessment (Farag et al., 2018; Mesgar and Strube, 2018), emphasize how to capture the global level of coherence in the text. Yang and Li (2021) proposed to exploit coherency sentiment representations to help implicit sentiment analysis. They focused on the local (word-level) aspect of sentiment coherency within the target sentence. Our ELCoM captures document-level coherence by using contrastive learning and sentence-level by hypergraph, and in this way, our model is sen-

sitive to capturing both global and local patterns, even in a small size of the training data such as the SemEval sentiment dataset.

## 3 Task Definition

Let $D = \{s_i\}_{i=1}^I$ be an input review consisting of the number of $I$ sentences, where the $i$-th sentence $s_i = \{w_j\}_{j=1}^n$ consists of the number of $n$ words. Let $a_{s_i} = \{a_{s_i}^t\}_{t=1}^T$ also be a set consisting of $T$ aspects within $s_i$. Each $a_{s_i}^t$ consists of a pair of the aspect-category and term. Let $c = \{$positive, negative, neutral$\}$ be a label set of sentiment polarities. The goal of the ABSA task is, for the given $t$-th aspect in $s_i$, to predict the sentiment label $l(a_{s_i}^t)$.

## 4 Approach

The overall architecture of the ELCoM is illustrated in Figure 2. It comprises three key steps: (1) representation learning with XLNet, (2) coherence modeling (CoM) to capture document-level and sentence-level coherence, and (3) cross-category enhancement to mitigate the influence of anomalous nodes.

### 4.1 Representation Learning with XLNet

We utilized XLNet (Yang et al., 2019) as the backbone model to obtain the sentence representation related to the target aspect and to review document representation. The XLNet is known to improve performance, especially for tasks involving a longer text sequence, e.g. text summarization, text classification, and text quality assessment. It is also utilized to model the coherence representation (Jwalapuram et al., 2022), because it takes advantage of both the autoregressive model and the BERT (Devlin et al., 2019).

Formally, for a sequence $\mathbf{x}$ of the length $I$, there are $I!$ possible orders for autoregressive factorization. Let $\mathcal{Z}_I$ be the set of total permutations of the length $I$ index sequence $[1,2,\cdots,I]$. $z_\tau$ and $\mathbf{z}_{<\tau}$ denote the $\tau$-th element and the first $\tau-1$ elements of a permutation $\mathbf{z} \in \mathcal{Z}_I$. The objective of XLNet is given by:

$$\max_\theta \mathbb{E}_{\mathbf{z}\sim\mathcal{Z}_I}\left[\sum_{\tau=1}^I \log p_\theta(x_{z_\tau}|\mathbf{x}_{\mathbf{z}_{<\tau}})\right]. \quad (1)$$

As such, the permutation language modeling optimization objective enables XLNet to effectively model the coherence across the document-level review.

Specifically, we create an input sentence, $s_i$ [SEP] $a_{s_i}^t$ [SEP] [CLS], for each aspect $a_{s_i}^t$ that appeared in the target sentence $s_i = \{w_j\}_{j=1}^n$. Here, the input is padded with two special symbols, [SEP] and [CLS], which are the same as those of BERT. We apply XLNet to the input and obtain each word embedding $\mathbf{e}_{w_j} \in \mathbb{R}^{d_m}$ and the aspect-based sentence embedding $\mathbf{e}_s \in \mathbb{R}^{d_m}$ marked with [CLS], where $d_m$ is the dimension size. Likewise, given the input document $D = \{s_i\}_{i=1}^I$, we concatenate it and create an input document sequence, $s_1$ [SEP] $\cdots$ $s_I$ [SEP] [CLS]. For the input sequence, we apply XLNet and obtain the document embedding $\mathbf{e}_d \in \mathbb{R}^{I\times d_m}$ marked with [CLS] that contains both document- and sentence-level representations.

### 4.2 Coherence Modeling

**Document-Level Coherence with Contrastive Learning.** Following Jwalapuram et al. (2022), to learn robust coherence representations, we adopt the sentence ordering task by using contrastive learning. It enforces that the coherence score of the positive sample (original document) should be higher than that of the negative sample (disorder document). Therefore, we disorder the original review to generate the number of $B$ negative samples by randomly shuffling the sentences within the document. For the results, we applied contrastive learning to align the coherent and incoherent representations. Let $f_\theta(\mathbf{e}_d)$ be a linear projection to convert coherent document embedding $\mathbf{e}_d$ into coherence scores. The margin-based contrastive loss is given by:

$$\mathcal{L}_{cl} = -\log(\frac{e^{f_\theta(\mathbf{e}_d{}^+)}}{e^{f_\theta(\mathbf{e}_d{}^+)} + \sum_{j=1}^B e^{(f_\theta(\mathbf{e}_{d_j}{}^-)-\tau)}}), \quad (2)$$

where $f_\theta(\mathbf{e}_d{}^+)$ indicates the coherence score of the positive sample, $f_\theta(\mathbf{e}_{d_1}{}^-)$, ..., $f_\theta(\mathbf{e}_{d_B}{}^-)$ denote the scores of $B$ negative samples, and $\tau$ is the margin. **Sentence-Level Coherence by Hypergraph.** Recall that more than 70% reviews have the same sentiment polarity if they have the same aspect-category. This indicates that the aspect-category is beneficial for sentiment identification. We thus utilize hypergraphs to exploit the relationships among sentences and aspect-categories. The hypergraph is a variant of the graph, where a hyperedge (edge in hypergraph) connects any number of vertices, while in graph-based methods, e.g., graph convolutional networks (GCN), an edge connects only two vertices (Yu and Qin, 2019; Wang et al., 2023).

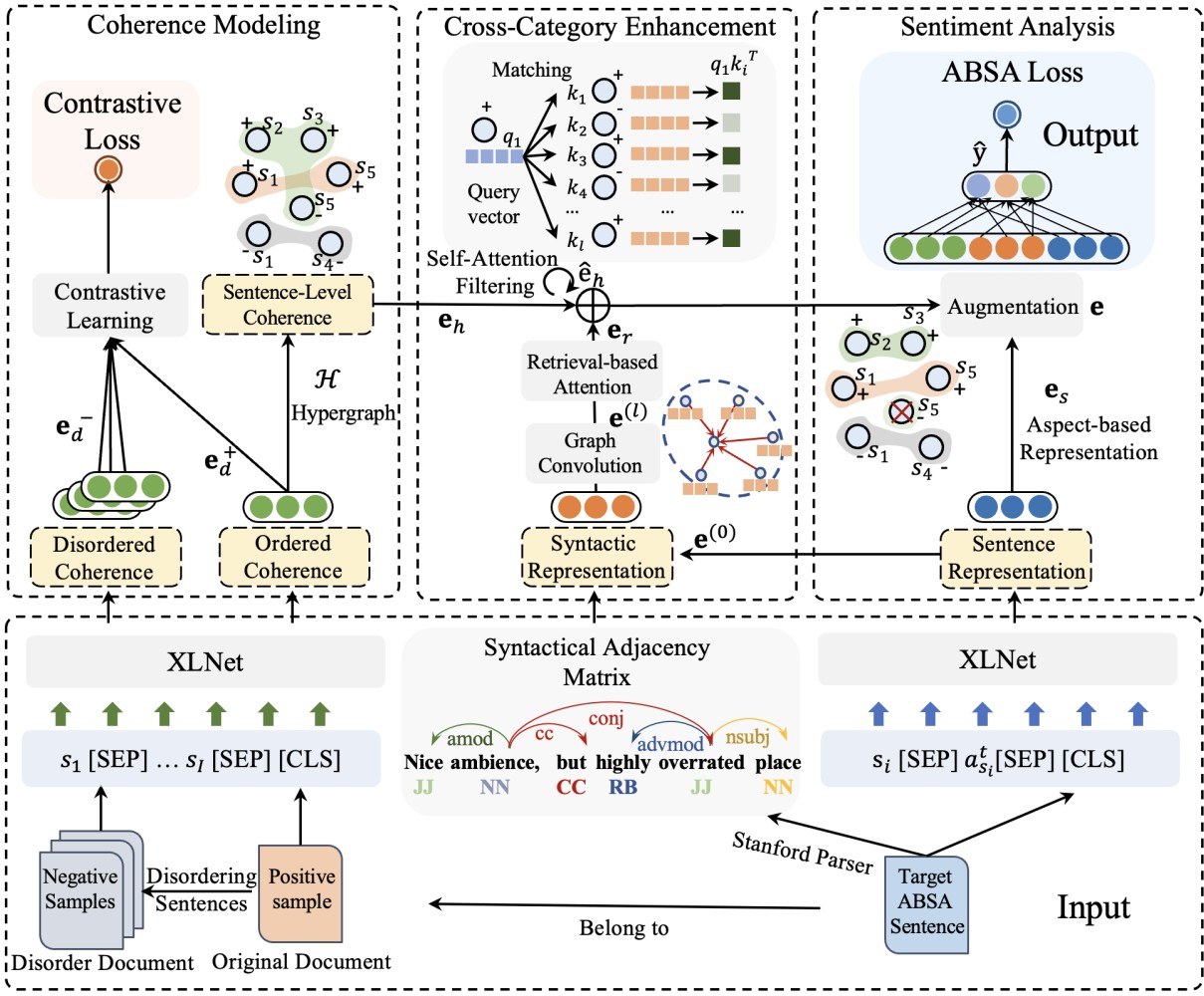

Figure 2: The architecture of the ELCoM. It comprises representation learning with XLNet, coherence modeling, and cross-category enhancement, with the input data consisting of original reviews and the target ABSA sentences. The final output is obtained through sentiment polarity classification.

We define aspect-categories ($C$ in total) as hyperedges, and sentences as nodes in the hypergraph $\mathcal{H} \in \mathbb{R}^{I \times C}$. Each row in $\mathcal{H}$ is for a sentence and each column is for the hyperedge of an aspect-category. $\mathcal{H}_{ij} = 1$ if vertex $i$ is connected by hyperedge $j$, and $\mathcal{H}_{ij} = 0$ otherwise. Note that the multiplication of the document embedding $\mathbf{e}_d$ and the set of multi-hot vectors ($\mathcal{H}\mathcal{H}^\top \in \mathbb{R}^{I \times I}$) can be regarded as the slicing operation that selects the corresponding embeddings of the sentences in the same category. The representation of the sentences toward the same aspect, $\mathbf{e}_h \in \mathbb{R}^{I \times d_m}$ is obtained by:

$$\mathbf{e}_h = \mathcal{H}\mathcal{H}^\top \mathbf{e}_d. \qquad (3)$$

### 4.3 Cross-Category Enhancement

We observed that 20~30% reviews contain different polarities even in the same category, while these reviews preserve coherence. The document- and sentence-level coherence modeling often leads to error propagation as it learns the information on both polarities from other sentences during training. To alleviate this issue, we perform cross-category enhancement on node embeddings in the hypergraph which is illustrated in Figure 2. More precisely, (1) we utilize a self-attention (SA) and reduce the influences between anomalous nodes, and (2) we apply a retrieval-based attention technique, Rba, to obtain enhanced embedding of the aspect term.

**Self-Attention (SA) Filtering.** Assume that if the aspect-category of the target query sentence contains different sentiments from the same aspect as other sentences, the SA weight should be small to dilute its features. We use the SA mechanism of the transformer, which is given by:

$$\mathbf{S} = \text{softmax}(\frac{\mathbf{Q}\mathbf{K}^\top}{\sqrt{d_k}})\mathbf{V}, \qquad (4)$$

where $\mathbf{Q}$, $\mathbf{K}$, and $\mathbf{V}$ refer to the query, key, and value matrices obtained by linear transformations of $\mathbf{e}_h$, respectively. The result is fed into a feedforward network, combined with layer normalization and residual connection. Each encoder layer takes the output of the previous layer as the input. This allows attention to be paid to all positions of the previous layer. The results are passed to an average pooling, and we obtain the filtered sentence representation $\hat{\mathbf{e}_h} \in \mathbb{R}^{d_m}$.

**Syntactic Representation Learning.** We use the Stanford parser[1] to obtain a dependency tree of the input sentence and apply graph convolution operations to learn the high-order correlations of the words. Given the syntactic adjacency matrix $\mathbf{A} \in \mathbb{R}^{n \times n}$ of the sentence and the embedding set of nodes (words) $\mathbf{e}^{(0)} = [\mathbf{e}_{w,1}, \mathbf{e}_{w,1}, ..., \mathbf{e}_{w,n}]$, the node representations are updated as follows:

$$\tilde{\mathbf{e}}_i^{(l)} = \sum_{j=1}^{n} \mathbf{A}_{ij} \mathbf{W}^{(l)} \mathbf{e}_j^{(l-1)},$$
$$\mathbf{e}_i^{(l)} = \text{ReLU}(\tilde{\mathbf{e}}_i^{(l)}/(d_i + 1) + \mathbf{b}^{(l)}), \qquad (5)$$

where $\mathbf{e}_j^{(l-1)} \in \mathbb{R}^{d_m}$ denotes the $j$-th word representation obtained from the GCN layer, $\mathbf{e}_i^{(l)}$ refers to the $i$-th word representation of the current GCN layer, and $d_i = \sum_{j=1}^{n} \mathbf{A}_{ij}$ is the degree of the $i$-th token in the dependency tree. The weights $\mathbf{W}^{(l)}$ and bias $\mathbf{b}^{(l)}$ are trainable parameters.

We apply Rba to the output of the GCN. Specifically, we mask out the non-aspect words in the output of the GCN to obtain the masked representation $\mathbf{e}_m$. The attention weights $\alpha_j$ of word $w_j$ are given by:

$$\alpha_j = \frac{\exp(\beta_j)}{\sum_{i=1}^{n} \exp(\beta_i)},$$
$$\beta_j = \sum_{i=1}^{n} (\mathbf{e}_{s_j})^\top \mathbf{e}_{m_i} = \sum_{i=\mu+1}^{\mu+m} (\mathbf{e}_{s_j})^\top \mathbf{e}_{m_i}, \qquad (6)$$

where the position of the target word ranges at $[\mu + 1, \mu + m]$, and $j \in [1, \mu+1) \cup (\mu+m, n]$ denotes the position of non-target words. $\beta_j$ calculates the semantic relatedness between the aspect and words other than the aspect in the sentence. The enhanced embedding of the aspect term is formulated by $\mathbf{e}_r = \sum_{j=1}^{n} \alpha_j \mathbf{e}_{s_j}$.

[1] https://stanfordnlp.github.io/CoreNLP/

| Dataset | Imp. | | Exp. | | COR (%) | COC (%) |
|---|---|---|---|---|---|---|
| | Train | Test | Train | Test | | |
| REST15 | 108 | 103 | 1560 | 736 | 57.43 | 78.29 |
| LAP15 | 132 | 4 | 1837 | 936 | 53.11 | 82.89 |
| REST16 | 211 | 41 | 2296 | 818 | 56.14 | 77.95 |
| LAP16 | 136 | 33 | 2773 | 768 | 53.01 | 81.39 |

Table 1: Statistics of the ABSA dataset. COR and COC indicate the ratio of reviews in which sentiment polarities are consistent, and in which sentiment polarities in the same aspect-category are consistent, respectively.

### 4.4 Multi-Task Learning

We use the multi-task learning (MTL) framework to optimize both the ABSA task and the sentence ordering task. As for the ABSA task, sentence representations with an enhanced aspect-category are obtained by $\mathbf{e} = [\mathbf{e}_s, \hat{\mathbf{e}}_h, \mathbf{e}_r]$. The result is passed on to the linear transformation layer. Using the softmax function, we obtain a probability score $\mathbf{p} \in \mathbb{R}^{|c|}$:

$$\mathbf{p} = \text{softmax}(\mathbf{W}_p \mathbf{e} + \mathbf{b}_p), \qquad (7)$$

where $\mathbf{W}_p \in \mathbb{R}^{|c| \times d_m}$ and $\mathbf{b}_p$ are the weight, and bias term, respectively. The task is trained with the cross-entropy loss, denoted as follows:

$$\mathcal{L}_{sa} = \sum_{I \in D} -(\gamma)^\top log(\mathbf{p}), \qquad (8)$$

where $\gamma \in \mathbb{R}^{|c|}$ denotes the true label vector.

Recall that we utilize margin-based contrastive loss $\mathcal{L}_{cl}$ to train the sentence ordering task. The final loss is given by:

$$\begin{aligned} \mathcal{L}_{(multi)}(\phi_{(sh)}, \phi_1, \phi_2) = \\ \delta_1 \mathcal{L}_{sa}(\phi_{(sh)}, \phi_1) + \delta_2 \mathcal{L}_{cl}(\phi_{(sh)}, \phi_2), \end{aligned} \qquad (9)$$

where $\phi_{(sh)}$ indicates the shared parameters, $\phi_1$ and $\phi_2$ stand for parameters estimated in the ABSA task and the sentence ordering task, respectively. $\delta_1, \delta_2 \in [0, 1]$ are hyperparameters used to balance the weights of the two tasks.

## 5 Experiments

### 5.1 Data and Evaluation Metrics

We conducted the experiments on four benchmark datasets: REST15 and LAP15 from the SemEval-2015 task12 (Pontiki et al., 2015), and REST16 and LAP16 from the SemEval-2016 task5 (Pontiki et al., 2016). The dataset consists of restaurant, and laptop domains, and positive, neutral, and negative

| Method | REST15 | | | | LAP15 | | | | REST16 | | | | LAP16 | | | |
|---|---|---|---|---|---|---|---|---|---|---|---|---|---|---|---|---|
| | ACC | F1 | IAC | EAC | ACC | F1 | IAC | EAC | ACC | F1 | IAC | EAC | ACC | F1 | IAC | EAC |
| ASGCN | 85.44 | 62.39 | 83.50 | 85.89 | 84.93 | 73.92 | **100.0** | 84.82 | 89.98 | 76.76 | 85.36 | 90.31 | 83.63 | 69.32 | 63.64 | 84.42 |
| Dual-GCN | 86.47 | 71.03 | 81.58 | 86.87 | 85.53 | 71.62 | **100.0** | 85.53 | 91.48 | 79.52 | 87.80 | 91.63 | 86.12 | 71.81 | 78.79 | 86.31 |
| AAGCN | 86.79 | 68.22 | 84.84 | 86.96 | 85.65 | 72.43 | **100.0** | 85.64 | 92.02 | 77.51 | 87.80 | 92.13 | 85.90 | 71.58 | 69.70 | 86.15 |
| Sentic-GCN | 85.89 | 70.67 | 84.54 | 86.09 | 85.82 | 72.89 | **100.0** | 85.87 | 91.23 | 79.31 | 85.36 | 91.37 | 85.27 | 71.61 | 69.70 | 85.78 |
| BiGAT # | 84.70 | 65.20 | - | - | 85.30 | 70.40 | - | - | 89.10 | 75.00 | - | - | 85.70 | 65.10 | - | - |
| MFGN | 86.92 | 68.54 | 80.62 | 87.41 | 85.94 | 72.18 | **100.0** | 85.88 | 91.89 | 80.91 | _92.69_ | 91.91 | 86.14 | 68.33 | 75.76 | 86.03 |
| SSEGCN | 87.08 | 69.07 | 81.51 | 86.93 | 85.58 | 72.11 | **100.0** | 85.54 | _92.13_ | 79.08 | 87.23 | _92.32_ | 85.64 | 70.78 | 78.79 | 85.91 |
| BERT-SPC | 85.92 | 66.02 | 78.58 | 86.89 | 85.68 | 74.32 | **100.0** | 85.64 | 90.89 | 78.11 | 90.24 | 91.37 | 84.07 | 68.09 | 75.76 | 84.27 |
| T-SCAPT | 85.32 | 66.31 | _86.44_ | 85.28 | 80.50 | 65.82 | **100.0** | 80.41 | 88.88 | 72.46 | 75.61 | 89.89 | 78.99 | 56.12 | 78.79 | 69.72 |
| B-SCAPT | _87.61_ | _71.88_ | 85.38 | _87.87_ | _88.96_ | _78.86_ | **100.0** | _89.02_ | 92.01 | 79.62 | 85.41 | 92.34 | 86.81 | 72.27 | _81.82_ | _86.88_ |
| CLEAN | 84.43 | 70.89 | 81.61 | 85.44 | 84.61 | 71.29 | **100.0** | 84.73 | 89.85 | 72.21 | 90.24 | 89.50 | 85.50 | 72.22 | 75.76 | 85.87 |
| CoGAN # | 84.20 | 70.70 | - | - | 85.10 | 74.50 | - | - | 92.00 | **81.60** | - | - | _87.20_ | _73.20_ | - | - |
| **w/o CoM** | 85.43 | 68.08 | 85.42 | 85.28 | 85.89 | 71.64 | **100.0** | 85.76 | 90.89 | 75.84 | 87.80 | 91.11 | 86.33 | 73.42 | 63.64 | 87.16 |
| **w/o CL** | 88.33 | 61.74 | 91.34 | 87.86 | 87.79 | 77.27 | **100.0** | 87.83 | 92.21 | 77.53 | 87.80 | 92.43 | 87.63 | 66.81 | 90.91 | 87.53 |
| **w/o CCE** | 87.08 | 67.23 | 89.29 | 86.83 | 87.42 | 76.54 | **100.0** | 87.27 | 91.91 | 76.22 | 92.69 | 91.83 | 87.94 | 62.54 | 87.88 | 87.93 |
| **ELCoM** | **89.63** | **72.53** | **92.23** | **89.34** | **89.52** | **79.71** | **100.0** | **89.23** | **93.36** | _81.32_ | **95.12** | **93.28** | **89.14** | **73.62** | **93.94** | **88.93** |

Table 2: Main results for four datasets. IAC and EAC refer to the accuracy of implicit and explicit sentences, respectively. "w/o CoM" refers to the result without any coherence information. # indicates the results from the original papers. Note that all the IACs of LAP15 are 100.0. The reason for this is that there are only four implicit sentiment sentences, and all baselines are identified correctly.

sentiment polarities. SemEval-2016 is labeled only with explicit sentiments. We thus manually annotated implicit sentiment labels in the dataset. The data statistics are shown in Table 1.

We used accuracy ACC (%), and macro-averaged F1 (%) scores as metrics. We evaluated our model by using implicit, and explicit sentiment accuracy, IAC (%), and EAC (%), respectively. For a fair evaluation, we conducted each experiment five times and reported the average results.

## 5.2 Implementation Details

Following Chen et al. (2020) and Cai et al. (2020), we randomly chose 10% of the training data and used it as the development data. The optimal hyper-parameters are as follows: The initial learning rate for coherence modeling was 6e-6 and others were 2e-5. The weight decay was set at 1e-3, and the dropout rate was 0.1. The number of negative samples $B$ was 5, and the margin $\tau$ was 0.1. The balance coefficients $\delta_1$ and $\delta_2$ were set at 0.9 and 0.1, respectively. The number of graph convolutional layers was 2. All hyperparameters were tuned using Optuna[2]. The search ranges are reported in Appendix A.3. We used AdamW (Loshchilov and Hutter, 2017) as the optimizer.

## 5.3 Baselines

We compared our approach with the following baselines:

[2]https://github.com/pfnet/optuna

1). **Graph neural networks (GNN)-based methods:** ASGCN (Zhang et al., 2019), Dual-GCN (Li et al., 2021a), AASGCN (Liang et al., 2021), Sentic-GCN (Liang et al., 2022), BiGAT (Shan et al., 2022), MGFN (Tang et al., 2022), and SSEGCN (Zhang et al., 2022);

2). **Knowledge-enhanced (KE)-based methods:** BERT-SPC (Song et al., 2019), TransEncAsp+SCAPT (T-SCAPT) (Li et al., 2021b), BERTAsp+SCAPT (B-SCAPT) (Li et al., 2021b), and CLEAN (Wang et al., 2022);

3). **Global context (GC)-based methods:** Co-GAN (Chen et al., 2020).

## 6 Results and Discussion

### 6.1 Performance Comparison

Table 2 shows the results. Overall, the ELCoM attained an improvement over the second-best methods by a 0.63~2.31% ACC and 0.57~1.08% F1-score, except for the F1-score on the REST16 dataset. In particular, it achieved remarkable results in implicit sentiment polarity classification, as the ELCoM achieved an improvement of IAC over the second-best method by 2.62~14.81%, while that of EAC was 0.24~2.36%. This reveals that leveraging document- and sentence-level coherence and reducing the influence of anomalous sentences significantly benefit sentiment analysis. Table 2 also provides the following observations and insights:

- Most baselines suffer from implicit sentiment analysis, while the ELCoM breaks the bottleneck

Textual review in REST15 <Review rid= P#9 >

**a1** **Raymond** the bartender rocks! **a2** **Pacifico** is a great place to casually hang out. The **a3** **drinks** are great, especially when made by **a4** **Raymond**.

The **a5** **omlette for brunch** is great... the **a6** **spinach** is fresh, definately not frozen... **a7** **guacamole** at pacifico is yummy, as are the **a8** **wings with**

**a9** **chimmichuri.** A weakness is the **chicken in the salads**. It's just average, **a10** **just shredded**, no seasoning on it. Also, I personally wasn't a

fan of the **a11** **portobello and asparagus mole**. Overall, decent **a12** **food** at a good **a13** **price**, with **a14** **friendly people**.

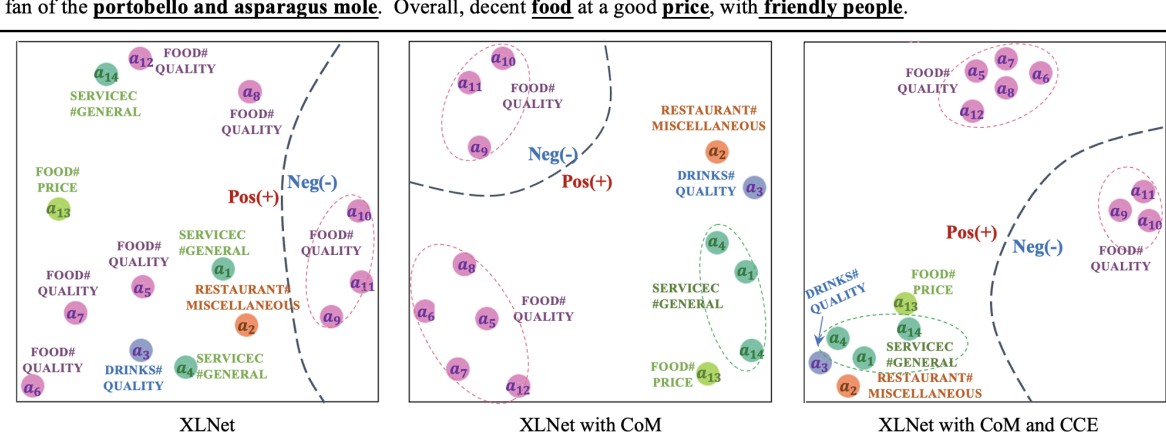

Figure 3: Visualization of the sentence representations. Each color corresponds to each aspect-category, and bold font underlined words refer to the aspect term. The dotted line separates positive and negative polarities.

and maintains good performance.

- SCAPT, which exploits pre-training on a large sentiment corpus and regards implicit and explicit sentiments as contrastive pairs, is competitive among baselines, and especially effective for the laptop domain. This indicates that prior knowledge is beneficial for sentiment analysis.
- GNN-based methods such as MGFN and SSEGCN achieved inspired results, suggesting that word-level syntactic representation can enrich sentiment features while ignoring that the reviewer's opinions are likely to be preserved throughout the review.
- The ELCoM and CoGAN exploit document-level sentiment knowledge, suggesting that capturing the reviewer's consistent sentiment expressions contributes to improving performance.

## 6.2 Ablation Study

We conducted an ablation study to examine the effects of each component of the ELCoM. The results in Table 2 prompts the following observations:

- The worst result, especially the ICA by the EL-CoM without applying CoM (w/o CoM) supports our hypothesis that the aspect-category is a strong clue for capturing sentence-level coherence to aid implicit sentiment classification.
- The ELCoM without contrastive learning (EL-CoM w/o CL), particularly the worst result on

| Category | | REST15 | | LAP15 | | REST16 | | LAP16 | |
|---|---|---|---|---|---|---|---|---|---|
| Sac | Rba | ACC | F1 | ACC | F1 | ACC | F1 | ACC | F1 |
| ✗ | ✗ | 87.1 | 67.2 | 87.4 | 76.5 | 91.9 | 76.2 | 87.9 | 62.5 |
| ✔ | ✗ | 88.3 | 70.1 | 88.7 | 78.9 | 92.6 | 77.8 | 88.4 | 66.3 |
| ✗ | ✔ | 87.3 | 69.0 | 88.3 | 77.3 | 92.9 | 79.8 | 89.0 | 67.8 |
| ✔ | ✔ | 89.6 | 72.5 | 89.5 | 79.7 | 93.4 | 81.3 | 89.1 | 73.6 |
| **Imp.(%)** | | 2.9 | 7.9 | 2.4 | 4.1 | 1.6 | 6.7 | 1.4 | 17.7 |

Table 3: Performance on cross-category enhancement. "Saf" indicates SA filtering, and "Rba" indicates retrieval-based attention. ✔ and ✗ denote with/without each module, respectively.

| Method | REST15 | | LAP15 | | REST16 | | LAP16 | |
|---|---|---|---|---|---|---|---|---|
| | ACC | F1 | ACC | F1 | ACC | F1 | ACC | F1 |
| SSEGCN | 81.9 | 62.5 | 80.1 | 67.3 | 83.9 | 74.9 | 78.6 | 64.4 |
| B-SCAPT | 82.1 | 61.7 | 83.7 | 72.7 | 83.3 | 70.5 | 79.1 | 62.5 |
| **w/o CCE** | 81.7 | 63.4 | 78.6 | 67.6 | 81.9 | 69.6 | 81.3 | 61.5 |
| **ELCoM** | 83.3 | 66.4 | 83.2 | 73.7 | 84.9 | 75.4 | 82.3 | 65.5 |

Table 4: Results on reviews that contain different polarities in the same category.

REST15 by F1, indicates that document-level coherence learned from the sentence ordering task contributes to improving performance.

- The ELCoM without cross-category enhancement (w/o CCE) suffers from a severe performance drop, particularly on LAP16 by F1, indicating that the SA mechanism that we used to reduce the influences between anomalous nodes is effective for accurate sentiment analysis.

| Case | Review Text. Underline: the target aspect to identify. | Aspect Category | BERT-SPC | B-SCAPT | CLEAN | **OURS** |
|---|---|---|---|---|---|---|
| 1 | $s_1$: Great services[pos]. Explicit 
 ... 
 (Implicit) 
 $s_4$: The **bus boy** even spotted that my table was shaking a stabilized it for me[pos] 
 $s_5$: **Food**[pos] was fine, with a some little-tastier-than-normal **salsa**[pos]. 
 Explicit ... Explicit | Bus boy (SERVICE) 
 Food (FOOD) 
 Salsa (FOOD) | Neg ✗ 
 Pos ✓ 
 Neg ✗ | Neg ✗ 
 Neu ✗ 
 Pos ✓ | Pos ✓ 
 Pos ✓ 
 Pos ✓ | Pos ✓ 
 Pos ✓ 
 Pos ✓ |
| 2 | ... 
 $s_4$: Maybe I wouldn't go back once more many years from now when I've forgotten I went there already[neg]. Explicit 
 $s_5$: Maybe I'll go back once more many years from now when I've forgotten I went there already[neg]. Implicit | (RESTAURANT) | Pos ✗ | Pos ✗ | Pos ✗ | Neg ✓ |
| 3 | $s_1$: No comparison[pos]. Implicit 
 $s_2$: I can't say enough about this place[pos]. Explicit 
 $s_3$: It has great sushi[pos] and even better service! Explicit | (RESTAURANT) | Neg ✗ | Neu ✗ | Neg ✗ | Pos ✓ |

Figure 4: Case study on REST16 data with their polarities predicted by BERT-SPC, B-SCAPT, CLEAN, and our approach: ✔ (or ✗) denotes that the predicted sentiment polarity is correct (or incorrect).

| Type of Contexts | REST15 | | LAP15 | | REST16 | | LAP16 | |
|---|---|---|---|---|---|---|---|---|
| | ACC | F1 | ACC | F1 | ACC | F1 | ACC | F1 |
| LSTM | 80.3 | 64.9 | 80.6 | 66.3 | 83.4 | 70.1 | 80.4 | 62.8 |
| SA | 87.8 | 66.9 | 87.6 | 74.6 | 91.4 | 75.2 | 87.9 | 67.2 |
| **Coherence** | **89.6** | **72.5** | **89.5** | **79.7** | **93.4** | **81.3** | **89.1** | **73.6** |

Table 5: Comparison against several network models.

Over-capturing contexts by SA.

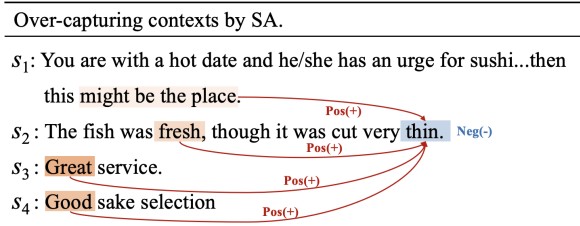

$s_1$: You are with a hot date and he/she has an urge for sushi...then this might be the place.

$s_2$: The fish was fresh, though it was cut very thin. Neg(-) / Pos(+)

$s_3$: Great service. Pos(+)

$s_4$: Good sake selection Pos(+)

Figure 5: Illustration of over-capturing contexts. Words such as "great," and "good" overly affect the negative opinion word "thin."

Recall that our cross-category enhancement utilizes two attention mechanisms, SA filtering, and Rba, to offset the impact of anomalous nodes. To examine the effectiveness of each mechanism, we performed experiments, which are shown in Table 3. Overall, the enhancement improved the performance, for instance, 1.36~2.93% by ACC and 4.14~17.72% by F1 in all datasets. Specifically, SA filtering yields more benefits for REST15 and LAP15, while Rba works well for other datasets.

It is interesting to note how cross-category enhancement dealing with anomalous nodes affects performance. Table 4 shows the results against SSEGCN and B-SCAPT by focusing on reviews containing opposite polarities in the same category. We can see that with CCE, the model improves 1.27~5.76% by ACC and 4.86~8.94% by F1. This clearly supports the effectiveness of our cross-category enhancement. To better understand the ablation study, we visualized the distribution of sentence representations of each module by t-SNE (Van der Maaten and Hinton, 2008), which is illustrated in Figure 3. We can see that (1) XLNet without CoM and cross-category enhancement roughly identifies the sentiment. $a_2$ is close to $a_9$, $a_{10}$, and $a_{11}$, although they have opposite polarities. (2) XLNet with CoM shows that the sentences that mention the same aspect are grouped together and share sentiments. (3) XLNet with CoM and cross-category enhancement shows that the dots in the same aspect-category are better clustered, while different aspect-categories are dispersed.

### 6.3 Efficacy of Coherence-Based Contexts

We compared the coherence-based contexts with the SA- and LSTM network-based ones to verify the effectiveness of the ELCoM, as these techniques are well known for effectively learning context dependencies. The results are shown in Table 5. It is reasonable that SA works better than LSTM, as the latter learns long-term dependencies. However, it reveals that SA may overcapture the context of irrelevant words even when the sentiment of the target aspect is explicit. As shown in Figure 5, $s_2$ contains two aspects: "food quality" and "food style." The SA-based context incorrectly predicts the sentiment polarity toward the latter aspect, as it overly captures positive sentiment from the words, "*great*," "*good*," and "*fresh*" as the descriptors of this aspect regardless of their dramatically effec-

tive transition, which is in fact not the case. In contrast, our approach captures not only the original sentiment in the sentence but also the context of sentiments. From the experimental results, we can conclude that coherence-based contexts are currently the optimal alternative compared to SA and LSTM in sentiment analysis tasks.

### 6.4 Case Study

We highlighted the typical and difficult examples and compared the ELCoM with the baselines. We chose BERT-SPC, B-SCAPT, and CLEAN as baselines, since BERT-SPC is often used as a benchmark model, and others are focused on implicit sentiment analysis. Figure 4 illustrates the results. **Case 1.** BERT-SPC and B-SCAPT failed implicit sentiment in $s_4$, as they could not correctly identify the sentiment of "service", which appears in $s_1$. Likewise, $s_5$ contains two aspects in the same category of "food", and because of its complex syntactic structure, it results in BERT-SPC incorrectly classifying the "salsa (food)" aspect as positive. To correctly identify these aspects, CLEAN infers causal representations of implicit sentiments in $s_1$ and $s_5$, and the ELCoM learns coherent contexts. **Case 2.** The sentiment of $s_5$ shows an implicit negative polarity in terms of the "restaurant". The ELCoM captures the polarity of the sentiment in $s_4$ via sentence-level coherence by hypergraph toward the same aspect-category to assist in sentiment classification. **Case 3.** It is extremely difficult to analyze the sentiment within short sentences, such as the *"No comparison"* in case 3. In contrast to the baselines, the ELCoM can capture the context of sentiment from $s_2$ and $s_3$ as auxiliary information.

### 6.5 Error Analysis on Explicit and Implicit Sentiments

We conducted error analyses on four datasets and found that the implicit sentence accuracy by EL-CoM is better than that of explicit sentences in some cases. There are three possible reasons:

- One reason is the effectiveness of the coherence modeling (CoM). Table 2 in Section 6.2 indicates that without CoM, the ELCoM obtains better accuracy on explicit sentiments than that of the implicit sentiments in the REST16 and LAP16 datasets.
- There are two major error cases on explicit sentences: (1) Many sentences containing neutral sentiments are incorrectly classified, because

neutral sentiments are always too ambiguous to identify, and neutral samples are not enough in training sets. (2) Many sentences with mixed (positive/negative) sentiment polarities were incorrectly identified, caused by negation words and unspecified referents within the sentences.

- The majority of implicit sentences do not have both positive and negative sentiment polarities, as a user often exerts an objective fact to express an implicit opinion, which means less probability of containing more than one different sentiment. In contrast, much more explicit sentences have mixed sentiment polarities. In the example from the REST16 test dataset which is shown in Figure 6, we can see that only one sentiment polarity, negative appears in the implicit sentence, while the sentence with explicit sentiment includes both positive and negative sentiment.

---

•**Explicit sentiment**: *I liked the atmosphere very much but the food was not worth the price.*
    category="ambience#general" polarity="positive"
    category="food#quality" polarity="negative"
    category="food#prices" polarity="negative"

•**Implicit sentiment**: *We are locals, and get the feeling the only way this place survives with such average food is because most customers are probably one-time customer tourists.*
    category="food#quality" polarity="negative"
    category="restaurant#general" polarity="negative"

---

Figure 6: Examples of explicit and implicit sentiments. Underlined words indicate aspect terms.

## 7 Conclusion

We proposed aspect-category enhanced learning with a neural coherence model (ELCoM) for implicit sentiment analysis. To mine opinions from explicit sentences to aid implicit sentiment classification, ELCoM captures document-level coherence by using contrastive learning, and sentence-level by a hypergraph. To further offset the impact of anomalous nodes in hyperedges, we proposed a cross-category enhancement on node embeddings. Extensive experiments have shown that the EL-CoM achieves competitive performance against state-of-the-art sentiment analysis methods. Future work includes, (i) improving the ELCoM by introducing a pre-training large sentiment corpus, and (ii) extending the ELCoM to simultaneously detect aspect-categories and their polarities (Cai et al., 2020).

## Limitations

The ELCoM model adopts time-consuming modules, i.e., transformer ($O(n^2)$) and GCN ($O(n^2)$) where $n$ refers to the number of words, therefore its computational cost heavily relies on the length of textual reviews. Although the ELCoM outperforms SOTA baselines in ACC, it still struggles with the performance in F1. One reason is because of neutral sentiments which are very ambiguous and difficult to predict correctly.

## Ethics Statement

This paper does not involve the presentation of a new dataset, an NLP application, and the utilization of demographic or identity characteristics information.

## Acknowledgements

We would like to thank anonymous reviewers for their thorough comments and suggestions. This work is supported by Kajima Foundation's Support Program. The first author is supported by the China Scholarship Council (No.202208330091).

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

# A Appendix

## A.1 Consistency of sentiment polarity.

Figure 7 shows the ratio of training and test reviews having consistent sentiment. We can see that 50.3%∼57.4% of the reviews had the same sentiment polarity regardless of different aspect categories over the review, and 70.8%∼84.8% of the reviews have the same sentiment polarity if they have the same aspect category.

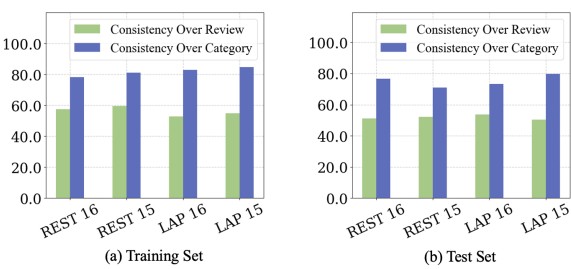

(a) Training Set      (b) Test Set

Figure 7: The ratio of reviews having consistent sentiment in different datasets. Consistency Over Review (COR): the ratio of reviews in which sentiment polarities are consistent. Consistency Over Category (COC): sentiment polarities in the same aspect category are consistent.

## A.2 Overview of coherence modeling and cross-category enhancement

Figure 8 illustrates a document- and sentence-level coherence and cross-category enhancement, where each circle alongside $s_i$ indicates a sentence in the document, and $a_i$ indicates the aspect-category. (a) of Figure 8 indicates document-level coherence, i.e., four sentences that contain negative sentiment with the same category indicate preserving coherence. (b) in Figure 8 shows sentence-level coherence. The sentiment of $s_1$ and $s_5$ toward the $a_1$ is enhanced by each other and avoids ineffective propagation from the sentiment in irrelevant aspect-categories, such as $s_4$ or the sentiment of $s_1$ related to the aspect-category $a_3$. In contrast, (c) of Figure 8 illustrates cross-category enhancement to compensate for the dilemma of side effects from the sentences that contain different polarities even in the same category. For example, in Figure 8 (c), the sentiment of $s_5$ related to aspect-category $a_2$ is prone to be incorrectly classified as positive, while the impact of $s_2$ and $s_3$ can be offset by cross-category enhancement.

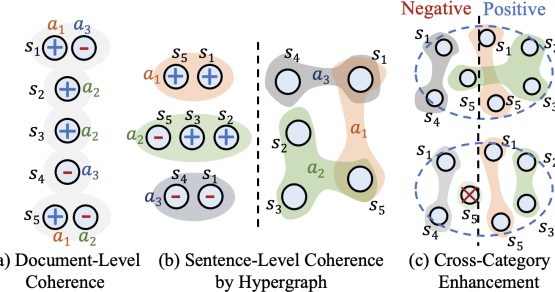

(a) Document-Level Coherence    (b) Sentence-Level Coherence by Hypergraph    (c) Cross-Category Enhancement

Figure 8: Illustration of a document- and sentence-level coherence, and cross-category enhancement.

## A.3 Implementation and hyperparameter setting

We implemented ELCoM and experimented with Pytorch on a single GPU: NVIDIA GeForce RTX 3090 (24GB memory). The search ranges of the hyperparameters used in our experiments are shown in Table 6.

| Parameter | Range |
|---|---|
| LR of CoM | 1e-6 $\sim$ 1e-5 |
| LR of others | 1e-5 $\sim$ 1e-4 |
| Weight decay | {1e-4, 1e-3, 1e-2} |
| Dropout rate | {0.1, 0.2, 0.3} |
| #Negative Samples $B$ | {5, 6, 7, 8, 9, 10} |
| Margin $\tau$ | {0.05, 0.1, 0.15, 0.2} |
| $\delta_1$ | {0.7, 0.8, 0.9, 1.0} |
| $\delta_2$ | {0.05, 0.1, 0.15, 0.2} |
| #Block of GCN | {1, 2, 3} |

Table 6: Search range of each hyperparameter: LR refers to the learning rate. LR of CoM indicates the learning rate of coherence modeling. LR of others shows aspect-based sentiment analysis and cross-category enhancement.

## A.4 Example of input data for multi-task learning

As shown in Table 7, the input data of the multi-task learning is as follows: the input of the sentence ordering task consists of the original review and its disordered review, and the input of the ABSA task comprises "Text" and "Opinions".

Following Li et al. (2021b), we labeled "Text" as implicit if it does not contain any obvious opinion words for a certain aspect. The number of $B$ disordered "Text" are generated from the original reviews by randomly shuffling the sentences, which is illustrated in Table 7.

| | Original Review [Review rid="1726473"] |
|---|---|
| Text | $s_1$: Average to good Thai food, but terrible delivery. |
| Opinions | target= "Thai food" category="food#quality" polarity="positive" implicit_sentiment="False" |
| | target="delivery" category="service#general" polarity="negative" implicit_sentiment="False" |
| Text | $s_2$: I've waited over one hour for food. |
| Opinions | target="null" category="service#general" polarity="negative" implicit_sentiment="False" |
| Text | $s_3$: They were very abrupt with me when I called and actually claimed the food was late because they were out of rice. |
| Opinions | target="null" category="service#general" polarity="negative" implicit_sentiment="False" |
| Text | $s_4$: A Thai restaurant out of rice during dinner? |
| Opinions | target="Thai restaurant" category="restaurant#miscellaneous" polarity="negative" implicit_sentiment="True" |
| Text | $s_5$: The food arrived 20 minutes after I called, cold and soggy. |
| Opinions | target="food" category="food#quality" polarity="negative" implicit_sentiment="False" |
| | target="null" category="service#general" polarity="negative" implicit_sentiment="True" |
| | Disordered Review |
| Text | $s_4$: A Thai restaurant out of rice during dinner? |
| Text | $s_1$: Average to good Thai food, but terrible delivery. |
| Text | $s_5$: The food arrived 20 minutes after I called, cold and soggy. |
| Text | $s_3$: They were very abrupt with me when I called and actually claimed the food was late because they were out of rice. |
| Text | $s_2$: I've waited over one hour for food. |

Table 7: The example of original and disordered review: "target" and "category" refer to aspect terms, and aspect category, respectively. Polarity takes a positive, neutral, or negative value. Implicit_sentiment takes "False" for explicit and "True" for the implicit sentiment.