# OpenReview forum: "Aspect-Category Enhanced Learning with a Neural Coherence Model for Implicit Sentiment Analysis"
_EMNLP/2023/Conference — EMNLP 2023 Findings_

### Official Review · Reviewer_3M1A · 2023-08-01

**Typos Grammar Style And Presentation Improvements:** 1. The abbreviation AC is first menti…
**Soundness:** 4

**Excitement:**

3: Ambivalent: It has merits (e.g., it reports state-of-the-art results, the idea is nice), but there are key weaknesses (e.g., it describes incremental work), and it can significantly benefit from another round of revision. However, I won't object to accepting it if my co-reviewers champion it.

**Paper Topic And Main Contributions:**

This paper tries to solve the Aspect-based Sentiment Analysis task. It proposes a novel neural coherence model named ELCoM to leverage the coherence of the article to mine the implicit aspect sentiments. The Coherence is measured at both the sentence and document levels, through hypergraphs and contrastive learning respectively. The idea is based on two observations made by the authors regarding sentiment relationships. An additional Cross-Category Enhancement module is designed to reset the nodes that do not follow the observed relationships.
The main contributions of the paper are:
1. The ELCoM model adopts coherence information for ABSA utilizing contrastive learning and hypergraphs.
2. The Cross-Category Enhancement (CCE) model to correctly offset the anomalous nodes.
3. Extensive experiments to evaluate the proposed model.

**Questions For The Authors:**

Question A: Regarding the hypergraph construction process in section 4.2, if two sentences have different sentiments on the same AC, will they still be connected by the same hyperedge?

**Reasons To Accept:**

The idea of leveraging document-level and sentence-level coherence for ABSA is very novel. The paper is easy to follow and the proposed model is well explained. Extensive experiments are conducted to verify the solidness of the model, accompanied with well-illustrated graphs.

**Reasons To Reject:**

1. As the paper is focused on implicit sentiment extraction, the authors should present the number and percentage of aspects with implicit sentiment in each dataset. In this way, the research purpose becomes more justified.
2. I notice that there are not any 2023 models for comparison. The authors should compare the proposed ELCoM to the most recent models.
3. MAMS is a novel and widely used dataset for ABSA evaluation. It will be better if ELCoM is evaluated and accessed on MAMS also.

**Reproducibility:**

4: Could mostly reproduce the results, but there may be some variation because of sample variance or minor variations in their interpretation of the protocol or method.

**Reviewer Confidence:**

5: Positive that my evaluation is correct. I read the paper very carefully and I am very familiar with related work.

---

> ### Author Rebuttal · Authors · 2023-08-28
>
> ### **Question A: Regarding the hypergraph construction process in section 4.2, if two sentences have different sentiments on the same AC, will they still be connected by the same hyperedge?**
>
> **A:** Yes, it is, and that is exactly why we performed a cross-category enhancement (CCE) to offset the impacts of anomalous nodes in the hypergraph. Specifically, we capture more fine-grained features (i.e., word-level syntactic representations via Standford Parser) and capture the correlations between the target sentence (to be classified) and the context sentences (which may mention the aspect category) by self-attention mechanisms for enhancement.
>
> ###  **Q: As the paper is focused on implicit sentiment extraction, the authors should present the number and percentage of aspects with implicit sentiment in each dataset. In this way, the research purpose becomes more justified.**
>
> **A:** Thank you for your helpful comments.
> The number of aspects (percentage) for implicit and explicit is 211/2296 (7.07%) in REST15, 136/2773 (4.18%) in LAP15, 252/3114 (6.64%) in RES16, and 169/3541 (1.42%) in LAP16, respectively. We will report these values in the paper.
>
> ###  **Q: I notice that there are not any 2023 models for comparison. The authors should compare the proposed ELCoM to the most recent models.**
>
> **A:** Thank you for your suggestion. We have followed the ACL 2023 conference and collected several related works [7, 8], although we are still waiting for them to release their codes for comparison. Once they have released their codes, we will try to add them.
>
> ###  **Q: MAMS is a novel and widely used dataset for ABSA evaluation. It will be better if ELCoM is evaluated and accessed on MAMS also.**
>
> **A:** Actually, we have investigated this dataset before our submission. Unfortunately, this dataset does not contain available document-level contexts, which cannot support our conjecture that the opinions from explicit sentences in the review can improve the performance of implicit sentiment analysis.
>
> ###  **Q: The abbreviation AC is first mentioned in the body of the paper in section 1, paragraph 2, but the meaning it stands for is not introduced until 2 paragraphs later. The explanation should be moved forward.**
>
> **A:** Thank you for pointing it out. We have moved the explanation of the abbreviation AC in section 1, paragraph 2 to section 1 line 51.
>
> ###  **Q: In section 5.1, paragraph 2, the evaluation metric for explicit sentiment accuracy should be ECA instead of EAC.**
>
> **A:** Thank you very much for your feedback. We have corrected the typo, and carefully checked our paper again.
>
> ###  **Reference:**
>
> [7] Ma F, Hu X, Liu A, et al. AMR-based Network for Aspect-based Sentiment Analysis[C]// ACL. 2023: 322-337.
> [8] Zhang M, Zhu Y, Liu Z, et al. Span-level Aspect-based Sentiment Analysis via Table Filling[C]// ACL. 2023: 9273-9284.

---

### Official Review · Reviewer_3JpN · 2023-08-03

**Soundness:** 4

**Excitement:**

4: Strong: This paper deepens the understanding of some phenomenon or lowers the barriers to an existing research direction.

**Paper Topic And Main Contributions:**

Implicit sentiment detection is a very challenging task in the aspect-based sentiment analysis applications. This paper proposes an aspect-category enhanced learning with a neural coherence model (ELCoM), which learns document-level coherence by using contrastive learning (CL) and sentence-level by hypergraph for mining opinions to aid implicit sentiment classification. Experiments on four benchmark datasets are provided.

**Questions For The Authors:**

a.	On some test datasets, IAC (implicit sentence accuracy) of the proposed method is better than EAC (explicit sentence accuracy). Why does not the proposed method work well on the explicit cases?  More illustration should be provided.

b.	In the evaluation，IAC and EAC refer to the accuracy of implicit and explicit sentences.  How to calculate IAC and EAC  for the sentences with multi aspects.

c.	How about the aspect-level accuracy of the proposed method?


**Reasons To Accept:**

Implicit sentiment detection is a very challenging task in the aspect-based sentiment analysis applications. This paper proposes a neural coherence model (ELCoM) for implicit sentiment detection, which captures document-level coherence contrastive learning (CL) and sentence-level by hypergraph in the sentiment analysis.

The proposed method is novelty and effective.

Extensive experiments on the popular datatsets demonstrate that the proposed method outperforms state-of-the-art sentiment analysis methods.

This paper is written and organized well.



**Reasons To Reject:**

 More experimental analysis should be provided. On some test datasets, IAC (implicit sentence accuracy) of the proposed method is better than EAC (explicit sentence accuracy). Why does not the proposed method work well on the explicit sentiment cases?  More illustration should be provided.

In the evaluation，IAC and EAC refer to the accuracy of implicit and explicit sentences.  How to calculate IAC and EAC  for the sentences with multiple aspects?

 Aspect-level evaluation should be reported since it is also very important.


**Reproducibility:**

4: Could mostly reproduce the results, but there may be some variation because of sample variance or minor variations in their interpretation of the protocol or method.

**Reviewer Confidence:**

5: Positive that my evaluation is correct. I read the paper very carefully and I am very familiar with related work.

---

> ### Author Rebuttal · Authors · 2023-08-28
>
> ### **Q: a. On some test datasets, IAC (implicit sentence accuracy) of the proposed method is better than EAC (explicit sentence accuracy). Why does not the proposed method work well on the explicit sentiment cases? More illustration should be provided.**
>
> **A:** Thank you for your comments and suggestions. In addition to the Case study section (Section 6.4), there are three possible reasons why IAC is better than EAC:
>
> 1). One reason is the effectiveness of the Coherence modeling (CoM). Table 2 in Section 6.2 shows that without CoM, the ELCoM obtains better accuracy on explicit sentiments than that of the implicit sentiments in the REST 16 and LAP 16 datasets. The accuracy of the IAC and EAC in REST15 is close. For the LAP 15 dataset, we cannot make a fair comparison because there are only four implicit sentences in this dataset.
>
> 2). We found two main error cases of explicit sentences: (1) Many sentences containing neutral sentiments are incorrectly classified, because neutral sentiments are always too ambiguous to identify, and neutral samples are not enough in training sets. (2) Many sentences with mixed (positive/negative) sentiment polarities were incorrectly identified, caused by negation words and unspecified referents within the sentences.
>
> 3). Furthermore, we examined datasets and found that the majority of implicit sentences do not have both positive and negative sentiment polarities, as a user often exerts an objective fact to express an implicit opinion, which means less probability of containing more than one different sentiment. In contrast, much more explicit sentences have mixed sentiment polarities. An example from the REST 16 test dataset is shown below. This could be one of the reasons for this result.
>
> - Example of explicit sentiment:
>
>   **S1: I liked the atmosphere very much but the food was not worth the price.**
>     target="atmosphere" category="ambience#general" polarity="positive"
>     target="food" category="food#quality" polarity="negative"
>     target="food" category="food#prices" polarity="negative"
>
> - Example of implicit sentiment:
>
>     **S2: We are locals, and get the feeling the only way this place survives with such average food is because most customers are probably one-time customer tourists.**
>     target="food" category="food#quality" polarity="negative"
>     target="place" category="restaurant#general" polarity="negative"
>
> We will mention these in our paper.
>
> ### **Q: b. In the evaluation，IAC and EAC refer to the accuracy of implicit and explicit sentences. How to calculate IAC and EAC for the sentences with multi aspects.**
>
> **A:**  An aspect denotes a pair of aspect category and term (line 166). In the experiments, we regard each aspect as one sample. For example, if there exist three aspects in a sentence, then there will be three samples for sentiment analysis. We used the same setting as [5] and [6].
>
> ### **Q: c. How about the aspect-level accuracy of the proposed method?**
> **A:** We are sorry for making you confused at this point. Our result is based on aspect-level accuracy.
>
> ### **Reference:**
>
> [5] Li Z, Zou Y, Zhang C, et al. Learning Implicit Sentiment in Aspect-based Sentiment Analysis with Supervised Contrastive Pre-Training[C]// EMNLP. 2021: 246-256.
> [6] Wang S, Zhou J, Sun C, et al. Causal Intervention Improves Implicit Sentiment Analysis[C]// COLING. 2022: 6966-6977.

---

### Official Review · Reviewer_mTUy · 2023-08-05

**Typos Grammar Style And Presentation Improvements:** 1. In line 200, \citet may be more pr…
**Soundness:** 2

**Excitement:**

3: Ambivalent: It has merits (e.g., it reports state-of-the-art results, the idea is nice), but there are key weaknesses (e.g., it describes incremental work), and it can significantly benefit from another round of revision. However, I won't object to accepting it if my co-reviewers champion it.

**Missing References:**

1. Yang, Heng, and Ke Li. "Improving Implicit Sentiment Learning via Local Sentiment Aggregation." arXiv preprint arXiv:2110.08604 (2021).

**Paper Topic And Main Contributions:**

The authors propose aspect category enhanced learning with a neural coherence model (ELCoM) to tackle the implicit sentiments in ABSA. The presented model can capture document- and sentence-level coherence by contrastive learning and hypergraph, respectively. Moreover, cross-category enhancements have been utilized to tackle sentiment conflict. The experimental results show that ELCoM achieves SOTA performance and reveal the effectiveness of ELCoM. The contribution are three folds: 1) propose ELCoM to aid implicit sentiment classification; 2) propose CCE to offset the impact of anomalous nodes; 3) extensive experiments and analysis on public datasets.

**Questions For The Authors:**

A. Why do you utilize XLNET as the representation model instead of BERT or RoBERTa?

B. How BERT and RoBERTa models perform in the framework of ELCoM?

C. What are the differences between your work and previous work that model aspect coherency? Related papers can be found in Missing References.


**Reasons To Accept:**

1. This paper is well-written.
2. Utilizing coherence model to recognize implicit sentiments is novel.
3. Detailed experiments not only show the ability of ELCoM but also give exhaustive analysis of the superiority of ELCoM.


**Reasons To Reject:**

1. It seems that there already have work done for implicit sentiment learning via aspect coherency, while the authors do not explain the differences.
2. Lack of explanation of utilizing XLNET as the representation model. Moreover, to provide a fair comparison, it is valuable to provide the results based on BERT or RoBERTa model.
3. Unfair comparison between baselines. Since most of the baselines are based on BERT and RoBERTa, it is unfair to compare these models with ELCoM based on XLNET.


**Reproducibility:**

4: Could mostly reproduce the results, but there may be some variation because of sample variance or minor variations in their interpretation of the protocol or method.

**Reviewer Confidence:**

3: Pretty sure, but there's a chance I missed something. Although I have a good feel for this area in general, I did not carefully check the paper's details, e.g., the math, experimental design, or novelty.

---

> ### Author Rebuttal · Authors · 2023-08-28
>
> ###  **Q: Why do you utilize XLNET as the representation model instead of BERT or RoBERTa?**
>
> **A:** Thank you for your feedback. There are three reasons why we utilized XLNet as a representation model:
>
> 1). XLNet uses permutation language modeling as the optimization objective. The XLNet is more appropriate for contrastive learning which aligns the coherent and incoherent representations (in Section 4.2) as it learns the correct order of all the tokens in the input.  It has been shown to be effective in capturing global dependencies between input tokens [1]. On the other hand, BERT and RoBERTa as masked language models focus on the contexts of masked words.
>
> 2). The max token length of both BERT and RoBERTa is 512. Therefore, many review texts must be manually deleted to be embedded by the BERT and RoBERTa models. There were 7.75% samples of REST 15 and 11.11% samples of LAP 16 datasets, leading to a lack of information in this task. In contrast, XLNet can handle long texts (i.e., 1,024), which is enough for all datasets.
>
> 3). Several recent studies have verified that XLNet is effective in modeling coherence tasks [2, 3] as XLNet can capture document (discourse) -level coherence.
>
> More interestingly, the ablation results in Table 2 (Section 6.2) show that even if we use XLNet as the backbone, ELCoM without coherence modeling (CoM) cannot exceed baselines except for F1 on LAP16, showing that the coherence model contributes most to improving the performance.
>
> We will mention the reason in our paper.
>
> ###  **Q: How BERT and RoBERTa models perform in the framework of ELCoM?**
>
> **A:** The table below shows the results (ACC) of ELCoM with XLNet, BERT, and RoBERTa. We only report ACC here due to space limitations, while we found that the results of IAC, EAC, and F1 had similar tendencies. The number of input tokens is the first 512 tokens for BERT and RoBERTa.
>
> |      |  REST 15 | LAP 15 | REST 16 | LAP 16 |
> | ----------- | ----------- | ----------- | ----------- | ----------- |
> | SOTA | 87.61 (**B-SCAPT**) | 88.96 (**B-SCAPT**)  |  92.13 (**SSEGCN**) | 87.20 (**CoGAN**)|
> | BERT | 86.30 | 87.16 | 90.56 | 86.62 |
> | RoBERTa | 87.02 | 89.06 | 92.32 | 87.75 |
> | XLNet | 89.63 | 89.52 | 93.36 | 89.14 |
>
>
> The results provide the following observations and insights:
>
> 1). We can see from the table that XLNet works better than BERT and RoBERTa. The result indicates that the XLNet is effective for modeling coherence.
>
> 2). ELCoM by BERT and RoBERTa are still comparable against the baselines by average -1.5%, and +0.1% improvement in ACC, respectively. Some of the results by BERT and RoBERTa in the framework of ELCoM were worse than the second-best results (underlined in Table 2, Section 6.1). This is reasonable because input token length is restricted, and they cannot model the document-level coherence which is one of the main contributions of our ELCoM.
>
> ### **Q: What are the differences between your work and previous work that model aspect coherency? Related papers can be found in Missing References**
>
> **A:** Thank you for your helpful suggestions. We will refer to Yang et al. work [4] and refine Section 2 Related Work to explain the differences between our work and [4].
>
> Yang et al. focused on the local (word-level) aspect of sentiment coherency within the target sentence [4]. They attempted to utilize three aspect features, i.e., sentence pair, local context-based, and syntactical local context-based to construct sentiment aggregation windows to model aspect sentiment coherency. Their approach is similar to ours in that they consider sentence-level information. The difference is that our model leverages opinions from explicit sentences by capturing both sentence- and document-level to aid implicit sentiment learning.
>
> ### **Q: In line 200, \citet may be more proper than \cite.**
>
> **A:** Thank you very much for your advice. We have corrected the citation in line 200, and carefully checked our paper again.
>
>
> ### **Reference:**
>
> [1] Yang Z, Dai Z, Yang Y, et al. Xlnet: Generalized autoregressive pretraining for language understanding[J]. Advances in neural information processing systems, 2019, 32.
> [2] Jwalapuram P, Joty S, Lin X. Rethinking Self-Supervision Objectives for Generalizable Coherence Modeling[C]// ACL. 2022: 6044-6059.
> [3] Jeon S, Strube M. Centering-based neural coherence modeling with hierarchical discourse segments[C]// EMNLP. 2020: 7458-7472.
> [4] Yang, Heng, and Ke Li. "Improving Implicit Sentiment Learning via Local Sentiment Aggregation." arXiv preprint arXiv:2110.08604 (2021).

---

### Meta-Review · Area_Chair_cfq3 · 2023-09-18

**Recommendation:** 4

**Metareview:**

There is a consensus among the reviewers that the paper presents a novel and well-organized approach to tackling implicit sentiments in ABSA. The proposed ELCoM model leverages both document-level and sentence-level coherence through contrastive learning and hypergraphs, and it also incorporates cross-category enhancements to address sentiment conflicts.
Reviewers have commended the paper for its clear presentation and extensive experiments, which demonstrate that ELCoM achieves state-of-the-art performance. However, there are also some concerns and suggestions for improvement. Reviewer 1, for instance, raises questions about the choice of the XLNET representation model over BERT or RoBERTa, and suggests a fairer comparison with these models. Reviewer 2 raises the issue of why ELCoM performs better on implicit sentence accuracy (IAC) than explicit sentence accuracy (EAC) on some test datasets and calls for more explanation. Reviewer 3 suggests that the paper should justify its research purpose by presenting the number and percentage of aspects with implicit sentiment in each dataset and also recommends evaluating ELCoM on the MAMS dataset.
In summary, the paper is well-received for its innovative approach to ABSA but also needs to address some concerns and incorporate additional experimental analysis to strengthen its claims.

---

### Decision · Program_Chairs · 2023-10-07

**Decision:**

Accept-Findings

**Comment:**

There is a consensus among the reviewers that the paper presents a novel and well-organized approach to tackling implicit sentiments in ABSA. The proposed ELCoM model leverages both document-level and sentence-level coherence through contrastive learning and hypergraphs, and it also incorporates cross-category enhancements to address sentiment conflicts.
Reviewers have commended the paper for its clear presentation and extensive experiments, which demonstrate that ELCoM achieves state-of-the-art performance. However, there are also some concerns and suggestions for improvement. Reviewer 1, for instance, raises questions about the choice of the XLNET representation model over BERT or RoBERTa, and suggests a fairer comparison with these models. Reviewer 2 raises the issue of why ELCoM performs better on implicit sentence accuracy (IAC) than explicit sentence accuracy (EAC) on some test datasets and calls for more explanation. Reviewer 3 suggests that the paper should justify its research purpose by presenting the number and percentage of aspects with implicit sentiment in each dataset and also recommends evaluating ELCoM on the MAMS dataset.
In summary, the paper is well-received for its innovative approach to ABSA but also needs to address some concerns and incorporate additional experimental analysis to strengthen its claims.